# Cadmium Stabilization and Redox Transformation Mechanism in Maize Using Nanoscale Zerovalent-Iron-Enriched Biochar in Cadmium-Contaminated Soil

**DOI:** 10.3390/plants11081074

**Published:** 2022-04-14

**Authors:** Sehar Razzaq, Beibei Zhou, Muhammad Zia-ur-Rehman, Muhammad Aamer Maqsood, Saddam Hussain, Ghous Bakhsh, Zhenshi Zhang, Qiang Yang, Adnan Raza Altaf

**Affiliations:** 1State Key Laboratory of Eco-Hydraulics in Northwest Arid Region of China, Xi’an University of Technology, Xi’an 710048, China; seharrazzaq688@yahoo.com; 2Institute of Soil & Environmental Sciences, University of Agriculture, Faisalabad 38040, Pakistan; ziasindhu1399@gmail.com (M.Z.-u.-R.); mohamgill@uaf.edu.pk (M.A.M.); 3Department of Agronomy, University of Agriculture, Faisalabad 38040, Pakistan; shussain@uaf.edu.pk; 4Training and Publicity, Agriculture Extension, Jaffarabad Balochistan, Dera Allah Yar 08289, Pakistan; ghouskhosa2353@gmail.com; 5Power China Northwest Engineering Corporation Limited, Xi’an 710065, China; zzs-483@nwh.cn (Z.Z.); yangqiang@nwh.cn (Q.Y.); 6School of Environmental Science and Engineering, Shandong University, Qingdao 266237, China; adnan.raza.altaf@gmail.com

**Keywords:** cadmium toxicity, Fe nanomaterial, antioxidants, Cd stabilization

## Abstract

Cadmium (Cd) is a readily available metal in the soil matrix, which obnoxiously affects plants and microbiota; thus, its removal has become a global concern. For this purpose, a multifunctional nanoscale zerovalent—iron enriched biochar (nZVI/BC) was used to alleviate the Cd—toxicity in maize. Results revealed that the nZVI/BC application significantly enhanced the plant growth (57%), chlorophyll contents (65%), intracellular permeability (61%), and biomass production index (76%) by restraining Cd uptake relative to Cd control. A Cd stabilization mechanism was proposed, suggesting that high dispersion of organic functional groups (C–O, C–N, Fe–O) over the surface of nZVI/BC might induce complex formations with cadmium by the ion exchange process. Besides this, the regular distribution and deep insertion of Fe particles in nZVI/BC prevent self-oxidation and over-accumulation of free radicals, which regulate the redox transformation by alleviating Cd/Fe^+^ translations in the plant. Current findings have exposed the diverse functions of nanoscale zerovalent-iron-enriched biochar on plant health and suggest that nZVI/BC is a competent material, feasible to control Cd hazards and improve crop growth and productivity in Cd-contaminated soil.

## 1. Introduction

Cadmium (Cd) is a common nephrotoxin with irreversible effects and is not safe to ingest at any level [1]. Thus, the Food and Drug Administration (FDA) has prescribed the maximum daily Cd intake level of 0.1 µg per kg body weight for chronic exposure [2]. Several sources of human exposure to Cd have passed this limit, including dietary, tobacco smoking, electroplating processes, industrial emissions, agrochemicals, and natural biogeochemical cycles [3]. Among other trace metals, Cd is highly mobile in soil solution and can easily translocate to plants, thereby threatening food security and crop yields [4,5]. Therefore, the World Health Organization (WHO) and the United States Environmental Protection Agency (US EPA) have set a safe limit of Cd concentrations (0.8 mg/kg: Agro–field area) [6,7] enacted to protect communities from Cd hazards.

To date, various sustainable and ecologically safe remediation strategies have been developed for metal-polluted soils [8,9,10,11]. Currently, the adsorption technique has been considered the most effective and economical approach for Cd contaminated soils [12,13]. In this regard, nanomaterials have gained prominence with excellent adsorption capacities and eco-friendliness [14,15,16]. Matin et al. [17] reported that nanomaterial interplay is a promising tool in absorbing and stabilizing the Cd in the soil matrix. Relevant studies have shown that nanomaterials’ application can considerably reduce the Cd availability and leachability in soils [18,19].

Among various nano-objects, nanoscale zerovalent iron material (nZVI) has made great progress in immobilizing metal ions and organic pollutants in water and soils [20,21]. It has been previously revealed that nZVI mitigates Cd toxicity in plants by generating an antioxidant defense system, improving plant physiology and soil fertility [22]. However, the self-oxidation by intracellular oxygen (Fe^0^ → Fe^2+^/Fe^3+^) and particle agglomeration reduces the reactivity of nZVI in soils [23]. In addition, the formations and leaching of Fe^2+^/Fe^3+^ ions may cause oxidative stress (ROS), leading to plant death [24]. Thus, preventing self-oxidation and particle aggregation in soil has a practical and scientific significance for the sustainable application of nZVI. Nonetheless, this issue can be solved by employing supporting materials to maintain the efficiency and ion dispersion of nZVI in the soil matrix. So far, many template materials have been used, such as zeolite, sepiolite, montmorillonite, attapulgite, graphite, activated carbon, and biochar [25].

Biochar is a reliable supporting material for the regular distribution of nZVI ions and reactivity due to a rich porous surface and high S_BET_ [26]. In addition, the high pH value coupled with plentiful surface functional groups (C=O, R–COOH, C(O)–O–C, –OH) and environmental friendliness make the biochar an ideal candidate for a Cd stabilization strategy within the matrix [27]. Until now, numerous studies have been focused on the removal of multiple pollutants from water using nZVI-loaded biochar [28,29]. Unfortunately, the effectiveness of nZVI-modified biochar as a reclaiming mediator in cultivated soils is still unknown, particularly on maize crops in Cd-polluted soil.

Therefore, this study comes with the aim to explore the impacts of nanoscale zerovalent-iron-loaded biochar (nZVI/BC) application on the growth, physiology, antioxidants regulation, cadmium uptake/translocation, and tolerance index in maize plants in Cd-contaminated soil. Moreover, possible Cd stabilization and redox transformation mechanisms were also discussed for scientific and practical reasons. This study will assist in using nZVI-loaded biochar for sustainable agriculture and reduce the risk of Cd toxicity to the environment.

## 2. Materials and Methods

### 2.1. Tea Waste Collection and Processing

Waste tea feedstock was used in biochar production due to natural abundance, having no commercial value, and sustainability. The methods of tea waste collection and sample processing are well defined in our previous work [30]. Briefly, waste tea leaves were collected from a nearby local tea stall and washed three times with distilled water. They were then naturally dried under good ventilation and sunlight for 24 h, finely ground, and sieved to obtain a 120–200 mesh size.

### 2.2. Material Synthesis and Characterization

The nZVI/BC was synthesized by a facile yet innovative one-step pyrolysis method [31]. Typically, 50 mL of 0.1 M Fe(NO_3_)_3_ solution (2.78 g) was prepared. Then, 5 g of processed waste tea sample was immersed in 0.1 M Fe(NO_3_)_3_ solution for 2 h under continuous thermostatic agitation (setup: 300 r/min at 80 °C). Subsequently, 50 mL of 0.75 M NaBH_4_ solution (2.83 g) was mixed dropwise to develop nZVI ions, followed by agitation. To maintain the reaction mixture pH (10–11), 10 M NaOH solution was used, with the process continued for 2 h. For high ion (Fe^0^) dispersion and more mass transfer at the liquid–solid interface, the solution was ultrasonically treated for 45 min at 45 °C [32]. Then, the nZVI loaded sample was separated and oven-dried for 8 h at 105 °C. For the optimum performance, the nZVI-loaded sample was pyrolyzed in a furnace at 500 °C for 2 h with a heating rate of 20 °C/min under pure N_2_ flow at 200 mL/min. Finally, the nZVI-loaded tea biochar (nZVI/BC) was obtained and kept in an airtight glass jar for further applications. The nZVI material for this study was synthesized by following the method described by Ravikumar et al. [33]. The synthesized materials, nZVI and nZVI/BC, were characterized by analytical techniques such as BET, FTIR, SEM, and XRD; detailed procedures are presented in the Appendix A. The tea feedstock without any chemical impregnation was used for ultimate and proximate analysis (Table 1).

### 2.3. Research Site and Experimental Conditions

The experiment was conducted under ambient temperature and light in the greenhouse of the ISES, University of Agriculture Faisalabad, Pakistan. The climate of the study area was humid and subtropical, with an average annual temperature of 35 °C and relative humidity of 45%. Before the experiment, an unpolluted soil sample was collected (20 cm depth) from the research area of ISES, UAF and analyzed per standard procedures followed by the ICARDA manual for the physicochemical characteristics (Table 2) [34].

For this study, treatments were arranged according to the completely randomized design by factorial and replicated thrice. A detailed experimental setup is presented in Table 3. The soil was artificially spiked with 100 mg/kg of Cd, using cadmium sulfate salt solution (CdSO_4_·8/3H_2_O) followed by spraying, and incubated for six weeks at 40% field capacity. After soil incubation, each pot was filled with 8 kg of soil. Then, nZVI and nZVI/BC were thoroughly mixed accordingly (Table 3) and incubated for 30 days. Finally, six sterilized seeds of hybrid maize (Monsanto 6317) were sowed in each pot, and two seedlings were retained in each pot after germination. Recommended doses per pot of nitrogen (N), phosphorous (P), and potassium (K) fertilizers were applied (Urea 0.75 g, DAP 0.87 g, SOP 0.64 g). Water irrigation regimes were applied, maintaining 70% field capacity throughout the experiment.

### 2.4. Pre-Harvest Analysis

Before crop harvesting (5 weeks after germination), plant physiological attributes including chlorophyll contents (mg g^−1^ FW), net photosynthetic rate (µmol CO_2_ m^−2^ s^−1^), transpiration rate (mmol H_2_O m^−2^ s^−1^), and stomatal conductance (mmol m^−2^ s^−1^) were recorded as per standard protocols described by Zhao et al. [35].

### 2.5. Post-Harvest Analysis

The maize plants were harvested 12 weeks after sowing, and plant growth parameters involving root length, plant height, root fresh weight, and shoot fresh weight were noted. Then, plant parts (roots and shoots) were separated and washed with tap water followed by distilled water, and oven-dried at 80 °C, to determine plant biomass production on a dry-weight basis.

### 2.6. Lipid Peroxidation Analysis

Malondialdehyde (MDA) contents were assayed to calculate the level of lipid peroxidation in the leaf. To prepare the assay mixture, a homogenized sample of the fresh leaf (0.25 g) and 0.5 mL of 0.1% trichloroacetic acid were centrifugated at 12,000× *g* for 15 min. Then, 4.0 mL of 20% trichloroacetic acid comprising 0.5% thiobarbituric acid was added to the aliquots of supernatants and heated at 95 °C for 15 min, then rapidly cooled in an ice-bath, then again centrifuged (12,000× *g*) for 5 min. The MDA contents were measured from the difference in absorbance (A532 nm–A600 nm) using Beer and Lambert’s equation [36].

### 2.7. Enzymatic and Non-Enzymatic Analysis

Antioxidant activities in leaves, including catalase (CAT), ascorbate peroxidase (APX), and superoxide dismutase (SOD), were estimated by spectrophotometry. For this purpose, ground plant leaf samples were pestled in liquid nitrogen and homogenized in phosphate buffer (M 0.05), maintaining pH at 7.8. Subsequently, the samples were filtered using four-layered muslin cloth and centrifuged for 10 min at 4 °C.

CAT activity was assayed according to Aebi’s method [37]; the assay solution (100 µL enzyme extract, 2.8 mL of 50 mM phosphate buffer with 2.0 mM citric acid (pH 7.0), and 100 µL of 300 mM H_2_O_2_) was prepared, and catalase activity was measured where H_2_O_2_ decomposed at 240 nm.

APX activity was determined by the method of Nakano and Asada [38]. An assay solution of 0.3 mL was obtained (100 µL enzymes extract, 2.7 mL of 50 mM potassium phosphate buffer with 2.0 mM ethylenediamine-tetra-acetic acid (pH 7.0), 100 µL of 300 mM H_2_O_2_, and 100 µL of 7.5 mM ascorbate), and ascorbate peroxidase activity was recorded through the reduction in absorbance at 290 nm due to decrease in ascorbate by H_2_O_2_.

SOD enzyme activity was determined by the photochemical reduction of Nitroblue Tetrazolium (NBT) for 3 min at 40 °C with a decrease in wavelength at 530 nm [39].

Proline contents were observed by a modified acid ninhydrin method [40]. For this purpose, 1 g leaf samples were homogenized with 3% sulphosalyclic acid and filtered. Then, ninhydrin and glacial acetic acid were added to the filtered mixtures and heated for 1 h at 100 °C in a water bath. In the last step, the toluene mixtures were extracted, and absorbance was taken at 520 nm.

### 2.8. Metal Uptake and Stabilization Analysis

The cadmium (Cd) concentrations in plant tissues (root and shoot) were determined by the atomic adsorption spectrophotometric technique. Typically, 50 mg of pulverized root and shoot samples were digested in 10 mL of HNO_3_: HClO_4_ mixture (3:1, *v*/*v*) and kept overnight. Subsequently, digested samples were heated at 350 °C until the solution became colorless. Finally, the Cd concentration in each precise solution was estimated using an atomic absorption spectrophotometer (AAS). The available Cd contents in post-harvested soil were determined using the AB-DTPA solution method [41]. The Cd uptake per plant part, total Cd uptake level, and Cd stabilization efficiency (η%) were calculated using the following Equations (1)–(3) [34].
(1)$$\mathrm{Cd}\;\mathrm{uptake}\;\left(\mathrm{ppm}\right)=\frac{\mathrm{Cd}\;\mathrm{in}\;\mathrm{extract}\;\left(\mathrm{mg}/\mathrm{kg}\right)-\;\mathrm{Cd}\;\mathrm{in}\;\mathrm{blank}}{\mathrm{Weight}\;\mathrm{of}\;\mathrm{soil}\;\left(g\right)}$$
(2)$$\mathrm{Total}\;\mathrm{Cd}\;\mathrm{uptake}\;\left(\mathrm{mg}/\mathrm{kg}\right)={\;\mathrm{Cd}}_{\mathrm{root}\;\mathrm{or}\;\mathrm{shoot}}\times \mathrm{respective}\;\mathrm{dry}\;\mathrm{biomass}\;\left(g\right)\;$$
(3)$$\eta \;\left(\%\right)=\frac{{\mathrm{Cd}}_{\mathrm{in}}-{\;\mathrm{Cd}}_{\mathrm{out}}}{{\mathrm{Cd}}_{\mathrm{in}}}\times 100\;$$

In Equations (1)–(3) Cd_root or shoot_ describes the Cd concentration in the root and shoot. The Cd_in_ and Cd_out_ represent the Cd contamination level (mg/kg) in the soil and total Cd uptake level (mg/kg) in the plant.

### 2.9. Bioconcentration Factor (BCF) and Translocation Factor (TF)

The bioconcentration factor was calculated to assess the metal uptake ratio from soil to the plant tissue (root or shoot). In addition, the translocation factor evaluates the capability of a phytoremediator species to translocate metal from the root to the ariel part (shoot or leaves) of the plant. The following Equations (4)–(6) were used to calculate the BCF and TF for this experiment.
(4)$$\mathrm{BCF}\;\mathrm{of}\;\mathrm{roots}\;=\frac{{\mathrm{Cd}}_{\mathrm{root}}\;\left(\mathrm{mg}/\mathrm{kg}\right)\;}{{\mathrm{Cd}}_{\mathrm{soil}}\;\left(\mathrm{mg}/\mathrm{kg}\right)}$$
(5)$$\mathrm{BCF}\;\mathrm{of}\;\mathrm{shoots}=\frac{{\mathrm{Cd}}_{\mathrm{shoot}}\;\left(\mathrm{mg}/\mathrm{kg}\right)\;}{{\mathrm{Cd}}_{\mathrm{soil}}\;\;\left(\mathrm{mg}/\mathrm{kg}\right)}\;$$
(6)$$\mathrm{TF}=\frac{{\mathrm{Cd}}_{\mathrm{shoot}}\left(\mathrm{mg}/\mathrm{kg}\right)}{{\mathrm{Cd}}_{\mathrm{root}}\;\;\left(\mathrm{mg}/\mathrm{kg}\right)}\;$$

In Equations (4)–(6) Cd_root_ and Cd_shoot_ describe the Cd concentration in the root and shoot.

### 2.10. Root/Shoot Ratio Factor (RF)

The root/shoot ratio is a critical measure that determines the metal stress confronted by the plants. The following Formula (7) describes estimating the root/shoot ratio factor [42].
(7)$$\mathrm{RF}=\frac{\mathrm{Root}\;\mathrm{dry}\;\mathrm{weight}\;\left(g\right)}{\mathrm{Shoot}\;\mathrm{dry}\;\mathrm{weight}\;\left(g\right)}\;$$

### 2.11. Plant Tolerance Lative Production Index Analysis

Plant tolerance index (PTI) and relative production index (RPI) factors were investigated to determine the influence of nZVI and nZVI/BC on dry matter production with/and without Cd contamination. Using the following Equations (4) and (5), plant tolerance and relative production index factors (%) were calculated.
(8)$$\mathrm{PTI}=\frac{R_{i}\;}{R_{0}}\times 100$$
(9)$$\mathrm{RPI}=\frac{{\mathrm{DM}}_{i}\;}{{\mathrm{DM}}_{0}}\times 100\;$$

In the above Equations (4) and (5), R_i_ and R_0_ represent the mean root length (cm) of the treated plant and mean root length (cm) of the control plant, respectively. DM_i_ and DM_0_ indicate dry matter production with Cd contamination and without Cd contamination, correspondingly.

### 2.12. Statistical Analysis

The maize plant data were determined by two-way analysis of variance (ANOVA) using data-processing software (Statistix, Vol. 8.1). The mean of three replications for each treatment were assessed by Tukey’s HSD post hoc test at (*p* ≤ 0.05). The graphs were developed using Origin software, version 9.1.

## 3. Results and Discussion

### 3.1. Characterization

#### 3.1.1. Brunauer–Emmett–Teller (BET) Analysis

Table 4 represents the textural characteristics of nZVI and nZVI/BC sorbents. Notably, nZVI showed a porous surface with a high BET surface area. However, a more uniform pore-size distribution with a prodigious specific surface area was revealed in nZVI/BC. This suggests that the Fe (NO_3_)_3_ saturation can promote pore formation by interacting with the sorbent surface during calcination. In addition, the intense ultrasonic irradiations produce various physio-chemical prodigies that could prohibit the aggregation of the particles [43].

#### 3.1.2. Scanning Electron Microscopy (SEM) Analysis

Similarly, the surface morphology of the synthesized materials was analyzed by scanning electron microscopy (SEM) and presented in Figure 1. Remarkable improvements in surface morphology with developed pore structures were yielded by nZVI/BC relative to the respective sorbent materials. As shown in Figure 1d, an excellent dispersion of nZVI NPs was observed over the surface of the nZVI/BC sorbent. However, particle agglomeration and pore obstruction were observed in the nZVI material. Hence, SEM evaluation encourages the BET results (Table 4).

#### 3.1.3. Fourier-Transform Infrared (FTIR) Analysis

FTIR analysis was performed to elucidate the concentration of organic functional groups on the synthesized materials. As shown in Figure 2, both sorbents revealed plentiful organic functional groups, including –OH at 3440 cm^−1^, C–H groups at 2747 cm^−1^, R–SH (thiol group) at 2554 cm^−1^, C=O (carboxylic, carbonyl, lactone, and ester) at 1562 cm^−1^, and Fe–O at 573 cm^−1^ [44]. However, the intensity of functional groups (–SH, –OH, C–O, C=O, and Fe–O) was relatively high on nZVI/BC.

#### 3.1.4. X-ray Diffraction (XRD) Analysis

Figure 3 reveals the XRD spectra of synthesized sorbents; major diffracting peaks in both nZVI and nZVI/BC at 47.6° were assigned to Fe^0^ [45]. However, two unique peaks indicative of iron oxides at 32.4° (Fe_3_O_4_) and 37.4° (γ–Fe_2_O_3_) were observed in nZVI/BC.

### 3.2. Impact on Plant Physiology

The plant physiological parameters, including chlorophyll contents, photosynthetic rate, transpiration rate, and stomatal conductance, were comparatively low in CdT. As shown in Figure 4a–d, the chlorophyll contents, photosynthetic rate, transpiration rate, and stomatal conductance were severely reduced by 21%, 71%, 126%, and 60% in CdT compared with the control. The elicited adverse effects on maize plant physiology are observed to be that Cd concentration beyond the permissible limit in soil (0.8–30 mg/kg) stimulates several phytotoxic features, leading to impeded plant health [8]. Observations that Cd toxicity reflects various metabolic, enzymatic, and genetic disorders in plants, and reduces water and nutrient assimilation have previously been described [46]; hence, plant physiology was affected.

All these physiological factors were significantly (*p* ≤ 0.05) improved by the exogenous application of nZVI in Cd–nZVI relative to CdT plants. Surprisingly, the plant physiological attributes were more effectively improved by nZVI/BC application in Cd-polluted soil compared with the other Cd-contaminated treatments. It can be seen in Figure 4a–e that remarkable improvements in chlorophyll contents (60% and 41.3%), photosynthetic rate (65% and 35%), transpiration rate (57% and 31%), and stomatal conductance (66% and 51%) were observed in Cd–nZVI/BC, respectively, in comparison with CdT and Cd–nZVI. Related studies have reported that iron (Fe) is an essential plant nutrient and constituent of several enzymes (Fds and Fd–NADP+) and pigments, which assist in restoring the photosynthetic e− transmission system, DNA synthesis, and gas exchange in abiotic stress [47]. In addition, biochar holds diverse crystalline structures of aromatic hydrocarbons (–COOH, –OH), responsible for absorbing toxic pollutants from the soil and protecting plant from abiotic stress [41]. The substantial improvements in maize plant physiology with Cd–nZVI/BC might be due to the above-discussed phenomenon.

### 3.3. Impact on Plant Growth

The plant growth attributes, including root length (RL), plant height (SL), root fresh weight (RFW), shoot fresh weight (SFW), root dry weight (RDW), and shoot dry weight (SDW) were considerably reduced in CdT by 31%, 62%, 49%, 51.2%, 30%, and 49.4%, correspondingly to controls, as displayed in Figure 5a–c. Cadmium is a non-essential element for plants and animals that brutally affects their metabolic activities [7]. Cd toxicity in plants may reduce their survival by impeding the Calvin cycle, and stimulating oxidative stress, lipid peroxidation, and plastoquinone production, ultimately reducing plant growth [48]. In Cd–nZVI, the RL, SL, RFW, SFW, RDW, and SDW were considerably (*p* ≤ 0.05) increased by 52%, 36%, 36.7%, 32%, 43%, and 19.4%, respectively, as compared with CdT. However, all plant growth parameters were more significantly improved with nZVI/BC application as shown in Figure 5c; the plant height improved by up to 67% and 56%, and plant dry biomass weight was enhanced by up to 57% and 33.4%, respectively, in Cd–nZVI/BC relative to CdT and Cd–nZVI. As in the above discussion, organic functional groups on the nZVI/BC surface may strengthen the Cd sorption efficiency and develop stable metal–organic complexes (Figure 2), limiting the Cd absorbing level in maize plants. Additionally, recent studies have shown that iron application, especially with nZVI, can boost biomass production by upregulating the plant defense system against abiotic stress; thus, plant growth attributes were considerably improved in Cd–nZVI/BC [45].

### 3.4. Impact on Lipid Peroxidation

Cadmium induces cellular injury upon binding with thiol and carbonyl groups (R–SH, C=O), resulting in replacement of essential cofactors that stimulate reactive oxygen species (ROS) [49]. In addition, the ROS species target the lipid molecules in the immediate vicinity, causing membrane damage (lipid peroxidation), eventually increasing malondialdehyde (MDA) contents [50]. The malondialdehyde production is a crucial sign of intracellular ROS discrepancy and lipid peroxidation [51]. Similarly, Figure 6a shows that CdT, Cd–nZVI, and Cd–nZVI/BC plants had a higher MDA ratio, indicating the imbalances of cell organelles due to overaccumulation of ROS under Cd contamination. However, the highest MDA contents were recorded in CdT maize plants. MDA contents are associated with oxidative stress (ROS), which impedes physiological processes and reduces plant growth [9]; thus, lower growth and physiology were observed in CdT. Surprisingly, significant differences were perceived in MDA contents between Cd–nZVI and Cd–nZVI/BC, which might have been due to active/passive supply of Fe contents to the plant. As revealed in Figure 6a, the MDA contents were comparatively higher in Cd–nZVI than Cd–nZVI/BC. Related studies have shown that nZVI holds active redox features and can self-oxidize (Fe^2+^/Fe^3+^) when reacting with oxygen and water, which might induce redox instability between the scavenging and yielding of ROS, resulting in overaccumulation of ROS via Fenton chemistry, thereby causing membrane lipid peroxidation [45,52]. These results are in good agreement with growth and physiological data; this is shown in Figure 4 and Figure 5. It can be further expounded by the following reactions (R1)–(R3).
Fe^0^ + O_2_ + 2H^+^ ⇌ H_2_O_2_ + Fe^2+^(R1)
(R2)$${\mathrm{Fe}}^{2+}+H_{2}O_{2}\;⇌\;\mathrm{OH}•+{\mathrm{Fe}}^{3+}+{\mathrm{OH}}^{-}$$
Fe^3+^ + O_2_ ⇌ O_2_•^−^ + Fe^3+^(R3)

Hence, we speculated that low MDA contents in CD–NZVI/BC might be due to the high dispersion of nZVI particles (Figure 1), and that intense surface organic functional groups (FTIR, Figure 2) generate more active sites for metal adsorption and maintain the standard redox potential. Several important enzymes such as catalase (CAT), superoxide dismutase (SOD), and ascorbate peroxidase (APX) have been identified to be indicators of free radical generation and oxidative stress in plants [53]. Therefore, enzymatic and non-enzymatic investigations were made to explain further and verify the above dynamic findings.

### 3.5. Impact on Antioxidants and Proline Modulations

Plants accumulate some non-toxic and highly soluble organic compounds in their bodies to combat and detoxify ROS effects. A significant inconsistency in antioxidant enzyme activity between Cd–nZVI and Cd–nZVI/BC was noticed. As shown in Figure 6b–e, catalase activity was higher by up to 11.4%, superoxide dismutase by up to 10.7%, and ascorbate peroxidase activity by up to 31% in Cd–nZVI as compared with Cd–nZVI/BC. Similarly, an amino acid test was explored to uncover the rapid recovery performance of synthesized materials against Cd-induced oxidative stress in maize plants. Proline is classified as a proteinogenic amino acid, playing an immense role in protecting the plant from various abiotic stresses and modulating stress recovery [22]. Apart from stabilizing the intracellular structures, proline improves plants’ agronomic parameters [54]. Likewise, proline contents were upregulated in Cd–nZVI compared with all other treatments, attributed to high stress-recovery performance with nZVI application in Cd-contaminated soil; this is shown in Figure 6e. Current findings verifying the above-discussed statements regarding high regulation of antioxidants (CAT, SOD, and APX) and proline activity indicate the presence of reactive species (O_2_•^−^, OH•, H_2_O_2_) which is an obvious feature of Cd toxicity, or might be the reason for an nZVI attack over protein and unsaturated fatty acids in phospholipids (R1–R3) [45].

However, the inconclusive enzymatic/non-enzymatic levels in Cd–nZVI/BC suggest that nZVI/BC could alleviate Cd toxicity in maize plants under certain conditions. It is believed that a high insertion of nZVI particles (Figure 3) developed various sulfur and other organic functional groups (–SH, C=O, –OH) on nZVI/BC, facilitating favorable conditions for Cd adsorption within the matrix [45]. Thus, it was hypothesized that adaptive hemostasis of enzymatic/non-enzymatic performance in Cd–nZVI/BC plants was probably due to the high spread and deep insertion on nZVI particles on biochar that ultimately prevented self-oxidation, particle agglomeration, and over-exploitation of iron particles, resulting in high Cd adsorption, and Cd stabilizing within the soil matrix in a non-toxic form. The current study suggests that nZVI-loaded biochar immobilizes the soil Cd and mediates the Fe/Cd-induced overaccumulation of reactive species in maize plants. To further validate these findings, metal uptake and translocation analyses were performed.

### 3.6. Impact on Cd Uptake and Stabilization Efficiency

The impacts on Cd assimilation and stabilization are presented in Table 5 and Figure 7. The Cd accumulation was comparatively higher at the rhizosphere to the aerial part of the maize plant. As shown in Figure 7, the maximum Cd uptake (root; 159.5 ± 3.25 ppm), bioaccumulation (root; 3.7 ± 0.93 ppm), and translocation (0.55 ± 0.52 ppm) were observed in CdT. Results revealed that with nZVI application, the Cd uptake (root), bioaccumulation (root), and translocation levels were significantly (*p* < 0.05) reduced by 43%, 17.2%, and 26.2%, respectively, compared with Cd–nZVI (Table 5). In comparison, the nZVI/BC application in Cd–nZVI/BC was found more effective in attenuating the Cd assimilation (root; 66% and 40%), bioaccumulation (root; 22% and 8.8%, shoot; 64% and 44%), and translocation (53% and 39.8%) in maize plants relative to CdT and Cd–nZVI. The low metal uptake and high stabilization efficiency eventually reduced the metal-stress ratio in plants; similar results were obtained in this experiment (Figure 7). These findings validate the cause of low MDA concentration and less enzymatic/non-enzymatic behaviors in Cd–nZVI/BC. Earlier investigations have shown that Fe is a multifunctional element with high redox potential and a larger surface area which facilitates adsorbing the Cd from the soil [55]. It has been reported that biochar application promotes the stable proportion of available toxic pollutants in the soil matrix, preventing their uptake by the plant [56]. Meanwhile, applying Fe-loaded biochar in soil may develop iron plaque on the plants’ root surface, limiting and/or reducing Cd translocation to plants’ root systems, thereby decreasing Cd assimilation by maize plants in Cd–nZVI/BC. These findings are in good agreement with SEM and XRD results, exhibiting the regular distribution of iron oxides over the porous biochar surface, favorable in capturing Cd from the soil [57]. It can be further explained by the following chemical reactions (R4)–(R8) [26].
(R4)$${\mathrm{Biochar}}_{\mathrm{surface}}─{\mathrm{COOH}}^{:}+{\;\mathrm{Cd}}^{2+}\;\overset{H_{2}O}{\to }\;\mathrm{Biochar}-{\mathrm{COOCd}}^{+}+2^{.}\mathrm{OH}+H^{+}\;$$
(R5)$${\mathrm{Biochar}}_{\mathrm{surface}}─{\mathrm{OH}}^{:}+{\;\mathrm{Cd}}^{2+}\overset{H_{2}O}{\to }\;\mathrm{Biochar}-{\mathrm{OCd}}^{+}+H_{3}^{+}O\;$$
(R6)$$\equiv {\mathrm{FeO}}^{-}+{\;\mathrm{Cd}}^{2+}\;\overset{H_{2}O}{\to }\;\equiv {\mathrm{FeOCd}}^{+}+\mathrm{OH}+H^{+}$$
(R7)$$\equiv \mathrm{FeOH}\;+{\;\mathrm{Cd}}^{2+}\;\overset{H_{2}O}{\to }\;\equiv \mathrm{FeOCdOH}+2H^{+}\;$$
(R8)$$2\mathrm{OH}+{\;\mathrm{Cd}}^{2+}\;\to \;\mathrm{Cd}{\left(\mathrm{OH}\right)}_{2}\;$$

### 3.7. Proposed Possible Cd Stabilization Mechanisms

Experimental and theoretical insights explained the possible Cd stabilization mechanisms on the surface of nZVI/BC. At first, it is proposed that under moisture conditions, the water-soluble organic compounds and minerals (especially Fe) might rapidly be dissoluted at the outer and inner surfaces on nZVI/BC, increasing dissolved organic carbon (DOC) and ion exchange capacity in the soil matrix. Secondly, after the rapid dissolution phase, the nZVI/BC might be coated (outside and inside) with a fine layer of multiple functional groups, especially C–O, C–N, Fe–O, and Fe–OH (Figure 2). This redox-active layer might have worked as a magnet for capturing/adsorbing various cations (especially heavy metals) anions, nano-objects, organic compounds, and minerals through a series of binding mechanisms (Figure 8), resulting in the formation of an organic mineral layer. Many studies have shown the O-functional groups on a biochar surface can actively immobilize the cationic metalloids, especially Cd^2+^ and Cu^2+^, through ion exchanges, precipitations, proton donation, and physisorption processes [58]. Hence, the Cd uptake, accumulation, and translocation were significantly reduced by the nZVI/BC application in Cd–nZVI/BC.

### 3.8. Impact on Plant Tolerance and Relative Production Index

The plant tolerance index (PTI) and relative production index (RPI) are indicative of crop productivity factors, and as previously reported, a plant with a higher tolerance index is considered to be more healthy, with optimum biomass production [59]. Results revealed that the PTI and RPI factors were exceptionally increased with nZVI/BC application in Cd-contaminated soil as well as in normal soil (Figure 9). The PTI factor of Cd–nZVI/BC reached about 65%, which was 33.2% and 13% higher, respectively, than CdT and Cd–nZVI in Cd-polluted soil. Similarly, admirable improvements in RPI factor (75%) were measured in Cd–nZVI/BC, approximately 44.8% and 32% raised with CdT and Cd–nZVI/BC. The present results verify the above research findings and descriptions of the growth, physiology, tolerance, metal-stress ratio, and Cd mediation in maize plants (Figure 4, Figure 5, Figure 6, Figure 7 and Figure 8). Moreover, it is conceived that nZVI/BC is a promising multifunctional candidate to enhance crop growth and productivity in a normal or stressful condition. Hence, the current study implies that nZVI/BC application can effectively mediate Cd toxicity and enhance plant growth and permissibility coupled with crop production under Cd stress and/or in normal conditions.

### 3.9. Environmental Significance and Limitations

The present work demonstrates that nZVI/BC is a multifunctional material capable of effectively mediating Cd toxicity in maize plants by regulating the hemostasis of Cd uptake. These findings may provide insight into the use of iron-assisted biochar material to remediate metal-polluted soils without compromising crop growth or human health. Since the soil texture used in this work was mainly sandy, other physical factors, including soil texture, annual precipitation, and soil microbes, might influence the Cd removal efficiency of nZVI/BC. Hence, further research is required to investigate Cd stabilization considering the aforementioned factors.

## 4. Conclusions

In this study, two multifunctional iron materials, nZVI and nZVI-assisted tea-biochar (nZVI/BC), were synthesized and used to alleviate the detrimental effects of cadmium toxicity on maize growth and physiology. Results revealed that photosynthetic rate, tolerance, and relative production index factor were severely reduced by 71%, 65%, and 75%, respectively, under Cd contamination (CdT). The nZVI application was effective, considering the plant physiological and agronomic parameters. However, nZVI/BC compendiously promoted the growth, physiology, and tolerance in maize plants by regulating the Cd homeostasis (92%) in soil. Thereby, it is concluded that nZVI/BC application might be a viable approach to improving crop growth and productivity without compromising human health in Cd-polluted soil.

## Figures and Tables

**Figure 1 plants-11-01074-f001:**
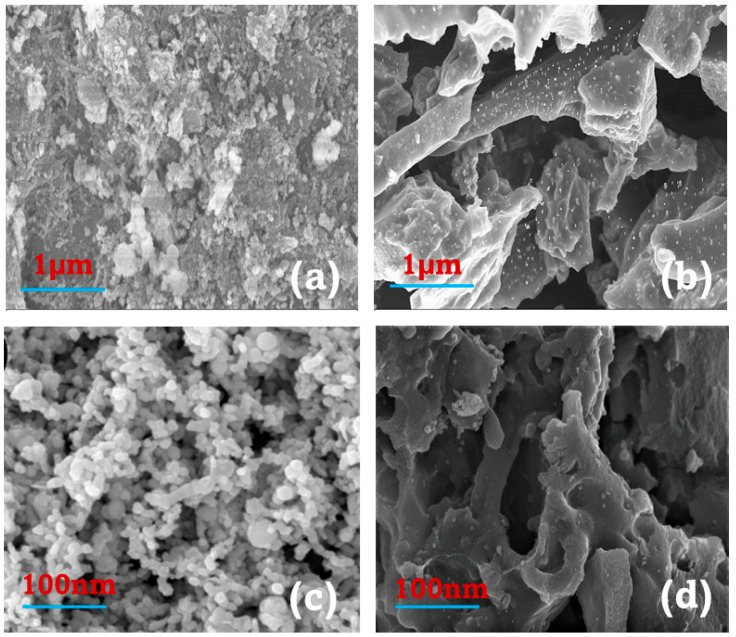
SEM results, (**a**,**c**) nanoscale zerovalent iron (nZVI); (**b**,**d**) nanoscale zerovalent-iron-loaded tea biochar (nZVI/BC).

**Figure 2 plants-11-01074-f002:**
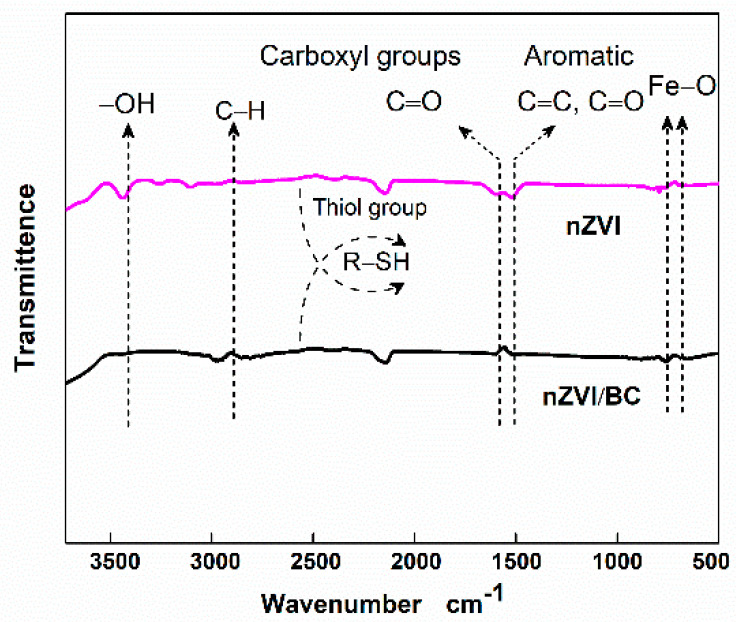
FTIR analysis of nZVI and nZVI/BC.

**Figure 3 plants-11-01074-f003:**
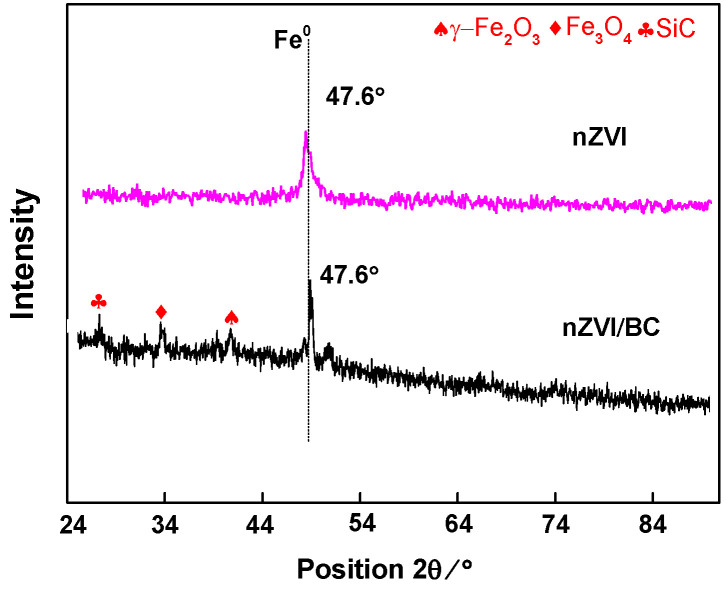
XRD analysis of nZVI and nZVI/BC.

**Figure 4 plants-11-01074-f004:**
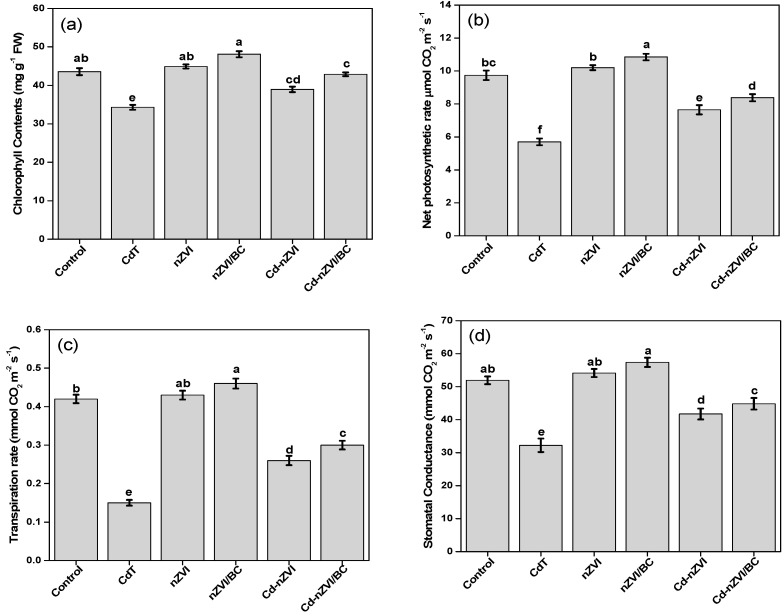
Impact of nZVI and nZVI/BC application on plant physiology in normal and Cd-polluted soils, (**a**) Chlorophyll contents; (**b**) Net photosynthetic rate; (**c**) Transpiration rate; (**d**) Stomatal conductance. Bars not sharing the same lower-case letter are significantly different (Tukey’s HSD α < 0.05). Represented data are the mean (*n* = 6) with ±standard deviation.

**Figure 5 plants-11-01074-f005:**
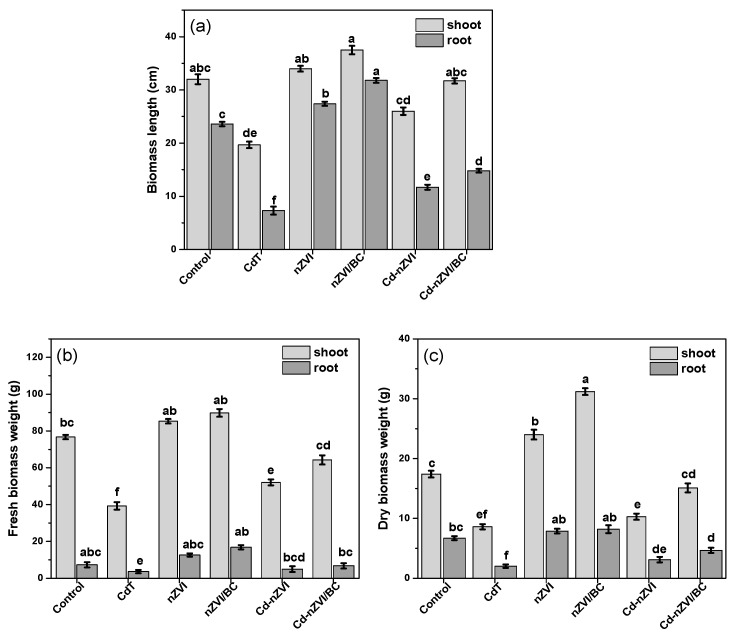
Impact of nZVI and nZVI/BC application on plant physiology in normal and Cd-polluted soils, (**a**) biomass length; (**b**) fresh biomass weight; (**c**) dry biomass weight. Bars not sharing the same lower-case letter are significantly different (Tukey’s HSD α < 0.05). Represented data are the means of three replications (*n* = 6) with ±standard deviation.

**Figure 6 plants-11-01074-f006:**
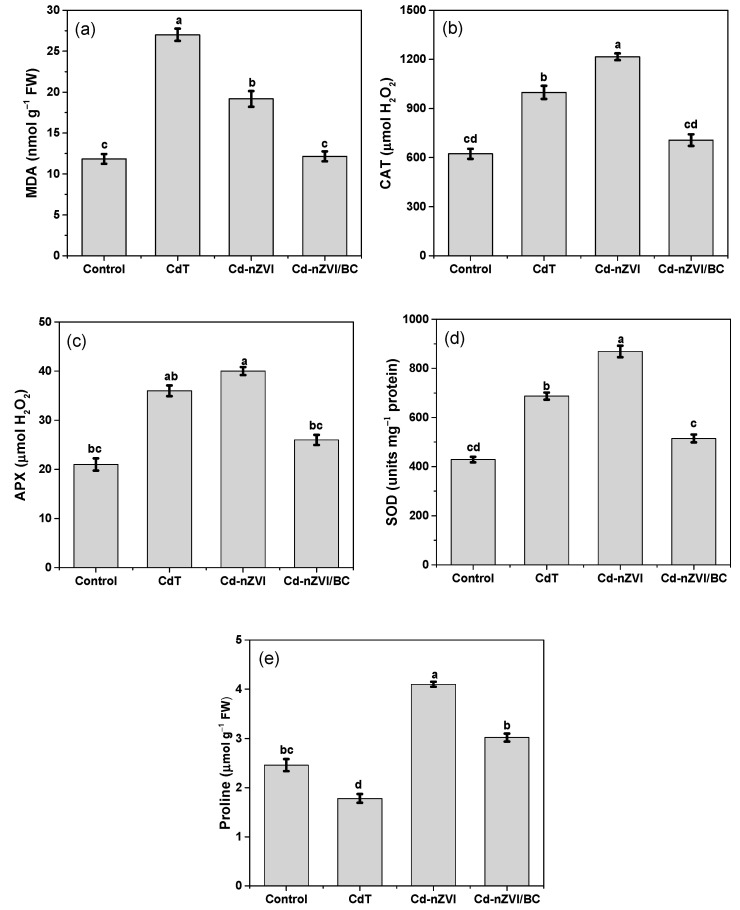
Impact of nZVI and nZVI/BC application on lipid peroxidation, enzymatic and non-enzymatic activities, (**a**) MDA contents; (**b**) catalase activity; (**c**) ascorbate peroxidase activity; (**d**) superoxide dismutase activity; and (**e**) proline modulation. Bars not sharing the same lower-case letter are significantly different (Tukey’s HSD α < 0.05). Represented data are the mean (*n* = 6) with ±standard deviation.

**Figure 7 plants-11-01074-f007:**
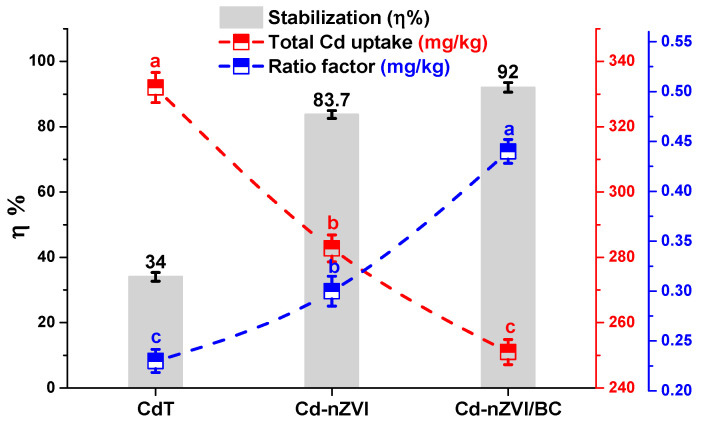
Impact of nZVI and nZVI/BC application in Cd stabilization performance, Total Cd uptake, and Root/shoot ratio factor. Bars not sharing the same lower-case letter are significantly different (Tukey’s HSD α < 0.05). Represented data are the mean (*n* = 6) with ±standard deviation.

**Figure 8 plants-11-01074-f008:**
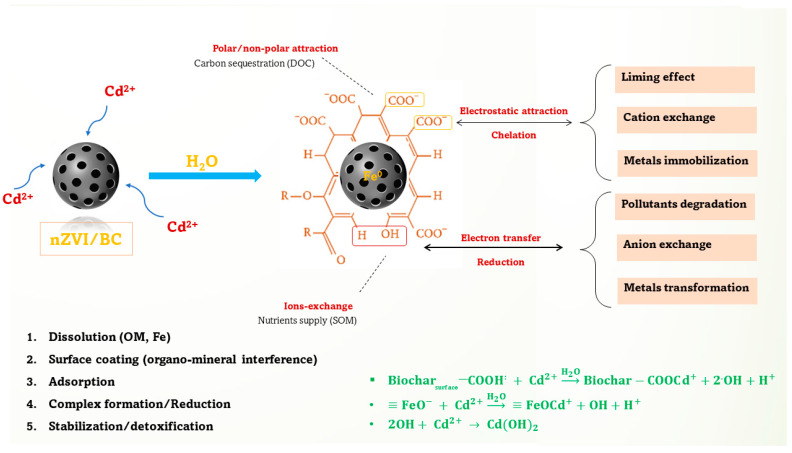
Possible Cd removal mechanisms on nZVI/BC.

**Figure 9 plants-11-01074-f009:**
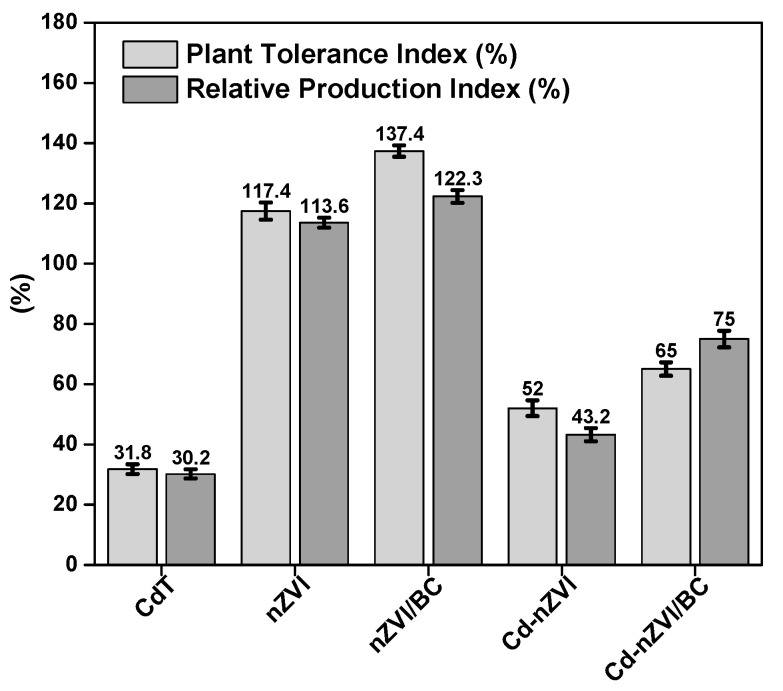
Impact of nZVI and nZVI/BC application on plant tolerance index and relative production index factors in normal and Cd-polluted soils. Bars not sharing the same lower-case letter are significantly different (Tukey’s HSD α < 0.05). Represented data are the means of three replications (*n* = 6) with ±standard deviation.

**Table 1 plants-11-01074-t001:** Proximate and Ultimate analysis of waste tea.

Proximate Analysis (%)				Ultimate Analysis (%)				
Ash	Volatile Matter	Moisture Contents	Fixed Carbon	C	H	O	N	S
5.21	78.43	1.86	14.37	63.70	4.69	33.26	4.532	0.36

**Table 2 plants-11-01074-t002:** Physico-chemical properties of potted soil.

pH	EC (dS m^−1^)	OM %	CEC Cmol_c_ kg^−1^	SS %	Textural Class	Sand %	Silt %	Clay %	N mg kg^−1^	P mg kg^−1^	K mg kg^−1^	Cd mg kg^−1^
7.67	3.6	0.66	1.31	38	Sandy clay loam	49.1	29.3	21.5	0.07	6.82	129	ND

OM = Organic Matter; SS = Soil Saturation; ND = Not Detected.

**Table 3 plants-11-01074-t003:** Experimental setup and description.

Treatments	Description
Control	With no amendment or contamination
CdT	Cadmium contamination level 100 mg/kg
nZVI	nZVI application 1%
nZVI/BC	nZVI/BC application 1%
Cd–nZVI	100 mg/kg Cd and 1% nZVI
Cd–nZVI/BC	100 mg/kg Cd and 1% nZVI/BC

**Table 4 plants-11-01074-t004:** The specific area and surface pore characteristics of prepared sorbents.

Samples	S_BET_ (m^2^/g)	Pore Volume (cm^3^/g)				Average Pore Width (nm)
		*V* _T_	*V* _mic_	*V* _meso_	*V*_%_ (*V*_meso_/*V*_T_)	
nZVI	141.42	0.491	0.253	0.374	77%	2.19
nZVI/BC	196.28	0.703	0.173	0.648	92%	1.42

**Table 5 plants-11-01074-t005:** Impact of nZVI and nZVI/BC application on Cd uptake and immobilization.

Treatments	Root (mg/kg per Pot)	Shoot (mg/kg per Pot)	Soil (mg/kg per Pot)	TF (mg/kg per Pot)	BCF Root (mg/kg per Pot)	BCF Shoot (mg/kg per Pot)
CdT	159.5 ± 3.13 a	87.8 ± 0.78 a	43.1 ± 1.46 a	0.55 ± 0.52 a	3.7 ± 0.93 a	2.03 ± 0.89 a
Cd–nZVI	91.2 ± 1.36 b	39.1 ± 0.7 b	29.8 ± 0.68 b	0.42 ± 0.14 a	3.06 ± 0.32 b	1.31 ± 0.77 b
Cd–nZVI/BC	54 ± 1.48 c	13.66 ± 0.9 c	18.7 ± 0.54 b	0.25 ± 0.46 b	2.88 ± 0.11 c	0.73 ± 0.32 c

Bars not sharing the same lower-case letter are significantly different (Tukey’s HSD α < 0.05). Represented data are the mean (*n* = 6) with ± standard deviation.

## Data Availability

All the data supporting this study are included in the article.

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
