# Peer review of "Cadmium Stabilization and Redox Transformation Mechanism in Maize Using Nanoscale Zerovalent-Iron-Enriched Biochar in Cadmium-Contaminated Soil"

_plants, 2022, doi:10.3390/plants11081074_

Round 1

Reviewer 1 Report

 Review Remarks on MS: entitled: ‘Cadmium–stabilization and redox transformation mechanism in maize using nanoscale zerovalent−iron enriched biochar in Cd contaminated soil. This study revealed application of nZVI/BC restricted Cd uptake which considerably improved plant growth and biomass output as compared to the control. The Cd–stabilization mechanism was postulated, implying that high dispersion of organic functional groups across the surface nZVI/BC might stimulate the formation of cadmium complexes via the ions exchange process.

The following are some general points and scientific queries that need to be addressed in the manuscript:

  • The language in the whole manuscript needs to be improved critically.  
  • In line 15, replace supported term with more appropriate one.
  • In line 17, physiology is not used in proper context, please provide results for individual parameters, or clearly state what do you mean by physiology because it is not a single a parameter.
  • Please pay sincere consideration for subscript and superscripts throughout the manuscript.
  • In line 38: US, EPA, please remove comma.
  • Please rephrase line 44-45.
  • Line 48, replace word capturing with suitable one.
  • In line 103, temperature is more appropriate than heat, pay attention.
  • In line 128, please provide the names of standard methods because these cannot be ignored.
  • Line 221: statistics software volume 8.1; please include the correct name of the software.
  • Please elaborate the term where they are used for first time throughout the manuscript.
  • In discussion, I rarely saw the supporting literature to support the current finding; why authors thought that their results are self-explanatory; please explain or revise discussion part according to the journal’s standard.
  • In each caption, means of three replications can be replaced with mean. Please mention the number of replications in statistical analysis.
  • Some less important Figure can be moved to the supplementary section.

Reviewer 2 Report

Extensive paper for scientists dealing with heavy metals in the soil and soil remediation. I think the most interesting is the part related to the mechanism, which explains how zero−valent iron-loaded biochar is reducing cadmium toxicity in plants. The research design and explanation of the results is correct… but I have some  comments:

Line 38, 87, 94, table 4 - choose one format for units (/ or -1)

Line 85 - ”2.78 g”…weight of  Fe(NO3)3 ???

Line 87, 91 -  ”…at 1 ± 80 ℃” ???...check it please

Line 97 - ”et al”…missing dot

Line 98 - BET, FTIR, SEM, and XRD - these terms should be defined the first time they appear (see the journal’s requirements)

Line 108 - please explain ”ICARDA”

Line 174 - ”AB-DTPA solution method”…any reference ???

Line 185, 186, 267, 385, table 5, E1,E2, E4-6 - ppm ??? … please use only SI units

Line 221 - ”statistics software volume 8.1”…what is this ???

Line 284 - explain ”Fds and Fe−NADP+”

Line 305, 340, 430 - ”(FTIR; Fig. 2)”… remove FTIR

Line 305 - ”Recent” start with a lower case letter

Line 339 - ”(SEM; Fig. 1)”… remove SEM

Line 347 - ”Lipids peroxidation” and ”Enzymatic and non-enzymatic activities” start with a lower case letter

Line 371 - ”(XRD; Fig. 3)” - remove XRD

Line 394 - ”(see Fig. 7)” - remove ”see”

Line 417 - ”Total Cd uptake”and ”Root/shoot ratio” start with a lower case letter

Line 432 - ”inions” ….ions ???

Line 433 - ”(see in Fig. 8)” - remove ”see in”

Table 1 - which methods were used in proximate and ultimate analysis ???; I think the terms ”proximate” and ”ultimate” are unfortunate…better remove

Table 3 - please correct the caption …move to the left

References

Check carefully all the references… please correct according to the journal’s requirements.

Round 2

Reviewer 1 Report

I am satisfied with the revised version, but I still have a few comments: Please elaborate the term "HSD" [i.e., honestly significant difference] where it was used for first time. Please check line 93:    for 45 min at 1 ± 45 ℃ Use one form, either FT-IR or FTIR. In line 308:  recent studies 308 have shown that iron application; I couldn't find references. Please check for typo errors throughout the manuscript.  
